# Evaluating Foundation Models' 3D Understanding Through Multi-View Correspondence Analysis

**Valentina Lilova**
University of Amsterdam
Amsterdam, 1098 XH
valentina.lilova@student.uva.nl

**Toyesh Chakravorty**
University of Amsterdam
Amsterdam, 1098 XH
toyesh.chakravorty@student.uva.nl

**Julian I. Bibo**
University of Amsterdam
Amsterdam, 1098 XH
julian.bibo@student.uva.nl

**Emma Boccaletti**
University of Amsterdam
Amsterdam, 1098 XH
emma.boccaletti@student.uva.nl

**Brandon Li**
University of Amsterdam
Amsterdam, 1098 XH
brandon.li@student.uva.nl

**Lívia Baxová**
University of Amsterdam
Amsterdam, 1098 XH
livia.baxova@student.uva.nl

**Cees Snoek**
University of Amsterdam
Amsterdam, 1098 XH
c.g.m.snoek@uva.nl

**Mohammadreza Salehi**
University of Amsterdam
Amsterdam, 1098 XH
s.salehidehnavi@uva.nl

## Abstract

This paper extends the Hummingbird framework with the Multi-View ImageNet (MVImgNet) dataset to evaluate how foundation model image encoders handle in-context object segmentation under unseen camera angles. We group MVImgNet object views and construct memory banks from selected viewpoints, assessing generalization by evaluating performance on held-out angles. In addition to seven pretrained Vision Transformer (ViT) models (CLIP, DINO, DINOv2, DINOv3, SigLIP2, C-RADIOv2, and TIPS), we include VGGT, a geometry-grounded ViT model trained for multi-view 3D scene understanding. Our results show that DINO-based encoders remain competitive across large viewpoint shifts, while 3D-aware models like VGGT require a dedicated multi-view implementation to properly reveal their geometric reasoning capabilities. These findings highlight the benefits of contrastive pretraining for robust performance across large viewpoint shifts. Our code is publicly available at https://github.com/ToyeshC/open-hummingbird-3d-eval.

## 1 Introduction

In recent years, we have seen the rise of foundation models [2, 25, 14, 30]. These large-scale pretrained models support many tasks, including visual understanding, which enables direct comparison of their encoders. Despite their impressive capabilities, state-of-the-art Vision Transformers (ViTs) can suffer catastrophic failures when objects are viewed from unusual angles, as demonstrated by recent angle-sensitivity studies [19]. However, most vision benchmarks emphasize single-view

tasks or dense synthesis [13], leaving segmentation robustness under camera rotations comparatively underexplored.

To address viewpoint variability, recent work has explored in-context learning (ICL) as a means to adapt models to unseen views without retraining. ICL is the ability of a model to perform new tasks by conditioning on a prompt [29]. An example is Hummingbird [1], a memory-augmented ViT for ICL. Its encoder extracts dense image features and projects them into key-value pairs stored in a dynamic memory. At query time, cross-attention over the keys assigns soft nearest-neighbor (NN) weights to aggregate values and predict novel views in-context. This modular design permits the use of any encoder.

In this work, we evaluate foundation model encoders for in-context object segmentation under unseen camera viewpoints. We extend the Hummingbird evaluation framework with the Multi-View ImageNet (MVImgNet) dataset [32], grouping object views into angular bins. By constructing memory banks consisting of images from specific viewpoints and evaluating segmentation performance on held-out angles, we assess how well different encoders generalize across viewpoint shifts. Our main contribution combines three elements: (1) pixel-level segmentation, (2) extreme viewpoint changes, and (3) dynamic memory for ICL. Most existing work studies 3D understanding, but there is little evaluation of how consistent the features from frozen 2D foundation models remain when the camera moves to new viewpoints, and only a few in-context examples are available.

In addition to general-purpose ViT encoders [16, 3, 15, 24, 26, 8, 12] we also include the Visual Geometry Grounded Transformer (VGGT) [27], a model designed for 3D scene understanding. Since VGGT does not provide native patch embeddings, we rely on its aggregated tokens to construct a token sequence compatible with Hummingbird. This provides a basis for comparing contrastive pretraining approaches with a geometry-grounded multi-view model.

## 2   Related work

Recent developments in view generation have significantly improved the understanding of 3D scene structure in computer vision, setting new benchmarks along the way. A notable example is Neural Radiance Fields (NeRF) [13], which achieves state-of-the-art novel view synthesis by optimizing a continuous volumetric function from sparse input images. While NeRF addresses generation, other studies investigate recognition in 3D understanding. Ruan et al. [19] demonstrate that modern recognition models (ResNets [6], ViTs [4], Swin [11], and Masked Autoencoders (MAEs) [7]) remain sensitive to viewpoint shifts in 3D scenes. With adversarial training, they improve invariance to 3D viewpoint changes beyond standard rotation-based augmentations. In parallel, Shifman and Weiss [23] show that state-of-the-art encoders such as CLIP and DINOv2 change predictions under a single-pixel shift in up to 40% of cases, despite extensive augmentation during training. Although their analysis centers on 2D translation invariance, the findings highlight that robustness to geometric changes, whether in 2D or 3D, remains limited in current vision encoders.

At a higher level, previous benchmarks in 3D computer vision were generally divided into two groups. The first group includes those working on traditional 3D recognition tasks, such as item detection (e.g., KITTI, Waymo [10, 9]) or scene segmentation (e.g., SAI3D, ScanNet [31]), which typically test explicit 3D architectures or use classification/bounding box tasks. The second focuses on innovative view synthesis approaches, such as NeRF [13], which prioritize dense creation rather than robust detection or segmentation under angular deviation. Beyond these 3D-focused benchmarks, general ICL benchmarks (such as the original Hummingbird framework [1] on PASCAL VOC [5]) assess feature transferability only in a 2D setting, and therefore fail to evaluate models under non-linear, wide-angle 3D viewpoint shifts.

Beyond model architectures, multi-view datasets such as MVImgNet and PASCAL3D+ [28] offer rich multi-view annotations that support the evaluation of cross-view generalization in object recognition. The rise of self-supervised Transformers has also opened new possibilities for downstream tasks such as segmentation. In particular, DINOv2 [3] produces robust embeddings that transfer effectively to prediction tasks, despite training without labels.

Nevertheless, a gap remains across these benchmarks and models. Most prior work emphasizes detection, classification, or view-angle estimation, but not the measurement of 3D contextual understanding through multi-view segmentation. To address this, our benchmark primarily evaluates

general-purpose foundation encoders, while also including a geometry-grounded model, VGGT [27], to compare against approaches explicitly designed for 3D scene understanding. To our knowledge, no existing benchmark systematically evaluates a model's ability to generalize segmentation across viewpoint shifts using dynamic memory. Our proposed evaluation method aims to address this gap in existing research.

# 3    Methodology

We evaluate the view generalization ability of frozen ViT models in semantic segmentation using a non-parametric, retrieval-based framework. Our approach builds on the Hummingbird framework [1], which applies ICL to vision tasks and supports the evaluation of spatial perception and semantic understanding. However, Hummingbird does not analyze performance variations across viewpoints changes. To address this, we introduce a viewpoint binning protocol and a multi-view dataset subset for cross-view robustness analysis. License information for the models, framework, and dataset can be found in Appendix A.

## 3.1    Models

We evaluate seven pretrained ViTs: CLIP [16], SigLIP2 [26], DINO [3], DINOv2 [15], DINOv3 [24], C-RADIOv2 [18, 8], and TIPS [12]. Most models use a ViT-B/16 backbone, except DINOv2 and TIPS, which have a ViT-B/14 backbone. Our benchmark primarily focuses on general-purpose foundation encoders, but for comparison we also include the geometry-aware VGGT [27]. This inclusion provides us a starting point to compare contrastive pretraining approaches and geometry-grounded representations.

## 3.2    Inference pipeline

We follow the Hummingbird [1] inference pipeline, where patch-level features from a frozen ViT are stored in a memory bank with one-hot semantic labels. During inference, query features are matched to the memory using FAISS with cosine similarity and $k = 30$ nearest neighbors. The memory size is passed as an input parameter. Hummingbird's cross-attention decoder aggregates the retrieved labels to produce segmentation predictions.

## 3.3    Dataset

Our study is based on MVImgNet, a large-scale dataset with over 6.5 million frames across 238 object categories [32]. Each frame includes a segmentation mask, camera extrinsics, and a reconstructed 3D point cloud. This dataset is chosen for three reasons: (1) it contains various annotations across wide viewpoint ranges, enabling view generalization analysis; (2) it includes camera extrinsics that allow precise angular binning; and (3) it has scale and diversity, capturing a broad range of object appearances and spatial configurations.

### 3.3.1    Viewpoint binning

To study viewpoint robustness, we discretize relative camera angles into seven bins spanning $0°$–$90°$, in steps of $15°$. To aid in this task, we use COLMAP [20–22]: a Structure-from-Motion (SfM) and Multi-View Stereo (MVS) pipeline that estimates camera poses and 3D scene geometry from a set of images. Using COLMAP extrinsics, we compute the relative rotation $R_{rel}$ between each frame and the first frame of the object instance. For each instance, we select one representative frame per bin by choosing the frame with the smallest angular error relative to the bin center. Images and masks are stored per bin for downstream use. The angular deviation $\theta$ is computed as

$$\theta = \arccos\left(\frac{\text{trace}(R_{rel}) - 1}{2}\right). \tag{1}$$

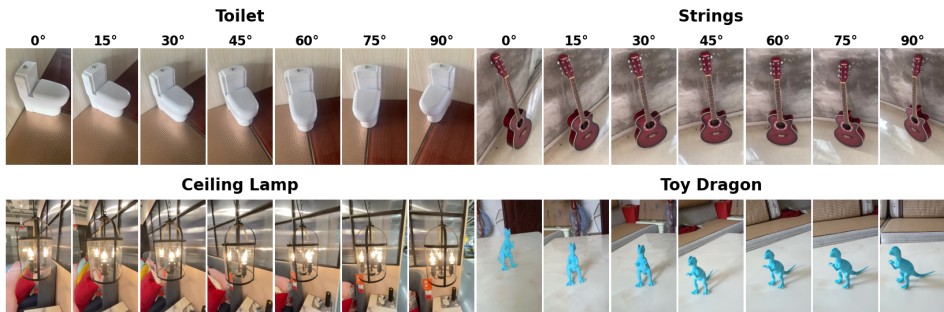

Figure 1: **Multi-view categories**. Each MVImgNet category is shown across all viewpoint bins (0°–90°), above are example visualizations for 4 of the 15 selected classes.

### 3.3.2 Subset construction

We curated a subset of MVImgNet, with the goal of selecting categories (classes) with sufficient angular coverage and manageable size for repeated memory-based evaluation. The two criteria used for selecting a category are: (1) the total zipped folder size must be 1–6 GB to ensure feasible data loading and storage; and (2) at least one object instance must span all seven angular bins with a maximum variance of 6°. Categories that did not meet these criteria were excluded. For example, category 23 (*laptop*) was excluded, as it did not have an instance that reached an angle of 90°. This yielded 15 segmentation classes, excluding the background.[1] Due to a misconfiguration in our codebase, eight classes had reduced per-bin image counts when evaluating the DINOv3 and VGGT models. All (complete and reduced) image counts per bin are found in Table 6 of Appendix B.

For each category instance, we parsed the COLMAP camera pose data to extract camera extrinsics and compute relative angles. For each valid instance, we selected the closest frame to each angular bin center and organized the resulting RGB images and masks in a structured directory grouped by category and angle. Additionally, we computed angular selection errors per bin (see Table 6). Examples of multi-view objects from the resulting dataset are shown in Figure 1.

### 3.4 Model input

To ensure a fair comparison, the largest native input size across the models is used: $504 \times 504$ for models that use a patch size of 14 and $512 \times 512$ for the ones using a patch size of 16. The evaluation of models with a bigger input size than their original resolution requires an additional step of embedding interpolation. In our setup, we upsample the absolute positional embeddings using bicubic interpolation to align with the new input resolution, as it provides smooth spatial transitions and is widely used for resizing in vision models. All evaluations are performed with a batch size of 4. The full configuration details are provided in Appendix C.

### 3.5 Metrics and multi-view evaluation

The segmentation quality is measured by mean Intersection over Union (mIoU) across all semantic classes. Each predicted segmentation mask consists of the background class and one of the 15 object classes. We report three metrics for each evaluation setting: (1) the per-class Intersection over Union (IoU), also known as the Jaccard index, (2) mIoU over all 16 classes, and (3) standard deviation. Results are presented per model, reference bin, and validation bin to analyze viewpoint robustness. Although all models perform well on the background class (see Figure 5), we include it in the mIoU computation for completeness.

---

[1]The 15 categories are ordered as follows: stove, sofa, microwave, bed, toy cat, toy cow, toy dragon, coat rack, guitar stand, ceiling lamp, toilet, sink, strings, broccoli and durian. More details can be found in Appendix B.

Table 1: **Difficulties and bin splits**. Difficulty levels are defined by the choice of reference bins (stored in the memory bank) and validation bins (unseen views for testing generalization). An increasing difficulty corresponds to fewer reference bins and larger unseen angular gaps, which we expect makes interpolation and extrapolation progressively harder.

| Difficulty | Reference bins | Validation bins |
|---|---|---|
| Easy | $0°, 30°, 60°, 90°$ | $15°, 45°, 75°$ |
| Medium | $0°, 45°, 90°$ | $15°, 30°, 60°, 75°$ |
| Hard | $0°, 90°$ | $15°, 30°, 45°, 60°, 75°$ |
| Extreme | $0°$ | $15°, 30°, 45°, 60°, 75°, 90°$ |

## 4 Experimental results

To validate our setup, we first reproduced the Hummingbird evaluation on PASCAL VOC [5] using the official open-source implementation.[2] The results, shown in Table 9 of Appendix E.1, match the methodology and configuration of the original paper [1]. This confirms that our pipeline is consistent with the benchmark and provides a reliable basis for the experiments that follow.

With the setup validated, we proceed to evaluate viewpoint generalization and robustness across three experiments: (A) cross-viewpoint generalization, (B) breaking point analysis, and (C) memory size robustness.

### 4.1 Experiment A: cross-viewpoint generalization

We compare the models in terms of their ability to generalize across viewpoints. The goal is to assess how different pretraining strategies affect performance when only limited reference views are available in the memory. Specifically, we evaluate how well the models generalize to unseen viewpoints when trained on selected angular bins from the MVImgNet dataset. The reference and validation splits vary across four difficulty levels, which are summarized in Table 1. Each model is evaluated using the mIoU across all 16 classes (background and 15 object categories) and all validation images. This setup enables a controlled comparison of each model's ability to interpolate between observed viewpoints and extrapolate to unseen ones.

#### 4.1.1 Results

As shown in Table 2, DINOv3 outperformed all models across all difficulties. DINO had the second-best performance for the Easy and Medium difficulty levels. Quantitative results illustrating these trends are shown in Figure 2, with per-class results in Appendix E.2. Across all classes, DINOv3, DINO, and DINOv2 had better segmentation performance under more difficult conditions than the other models. Overall, CLIP consistently followed close behind the DINO models, while C-RADIOv2, TIPS, and SigLIP2 performed noticeably worse, especially under limited reference view conditions. VGGT showed the lowest scores in our setting.

#### 4.1.2 Discussion

Although none of the models were explicitly trained for viewpoint understanding, the results show clear differences in their ability to generalize across views. As seen in the previous section, DINOv3 performs best, followed by DINOv2 and DINO. This indicates that the features of the DINO models, compared to other models, capture object shape more reliably and remain consistent under viewpoint changes. This consistency may arises from their self-supervised training objective, which aligns features across different augmentations of the same image and promotes stable relations between local and global cues, leading to structured representations where similar object parts share similar embeddings. DINOv3 further benefits from large-scale pretraining and the Gram Anchoring mechanism [24], leading to more stable local features and stronger geometric consistency under viewpoint changes.

While DINO performed better under the easier difficulties, where multiple views of the object were available, DINOv2 obtained a higher mIoU under the Extreme difficulty with only one reference

---

[2]The evaluation repository can be found at `https://github.com/vpariza/open-hummingbird-eval`.

Table 2: **mIoU scores across difficulty levels**. For each model, we report the average mIoU and standard deviation over the four difficulty levels. The memory size used is 1,024k. DINOv3 achieves the best performance across all difficulties, followed by DINOv2 and DINO, of which the latter outperformed DINOv2 in the Easy and Medium setups (where multiple reference views are available). DINOv2 significantly outperformed DINO under the Extreme case with only a single reference bin. VGGT shows significantly worse results than other models.

| Model | Easy | Medium | Hard | Extreme |
|---|---|---|---|---|
| CLIP ViT-B/16 | $0.755 \pm 0.130$ | $0.748 \pm 0.134$ | $0.734 \pm 0.140$ | $0.701 \pm 0.149$ |
| DINO ViT-B/16 | $0.782 \pm 0.132$ | $0.774 \pm 0.137$ | $0.748 \pm 0.153$ | $0.686 \pm 0.171$ |
| DINOv2 ViT-B/14 | $0.763 \pm 0.136$ | $0.758 \pm 0.139$ | $0.748 \pm 0.143$ | $0.728 \pm 0.154$ |
| DINOv3 ViT-B/16 | $\mathbf{0.807 \pm 0.153}$ | $\mathbf{0.805 \pm 0.146}$ | $\mathbf{0.794 \pm 0.161}$ | $\mathbf{0.766 \pm 0.191}$ |
| C-RADIOv2 ViT-B/16-CPE | $0.653 \pm 0.129$ | $0.636 \pm 0.132$ | $0.592 \pm 0.141$ | $0.506 \pm 0.150$ |
| SigLIP 2 B/16-512 | $0.564 \pm 0.150$ | $0.551 \pm 0.153$ | $0.530 \pm 0.157$ | $0.481 \pm 0.152$ |
| TIPS ViT-B/14-HR | $0.667 \pm 0.137$ | $0.647 \pm 0.146$ | $0.588 \pm 0.162$ | $0.462 \pm 0.169$ |
| VGGT | $0.190 \pm 0.132$ | $0.172 \pm 0.119$ | $0.146 \pm 0.102$ | $0.112 \pm 0.083$ |

bin. This result suggests that DINO's simpler self-distillation loss favors interpolation across views, whereas DINOv2's larger-scale self-supervised pretraining supports stronger extrapolation to unseen angles. CLIP generally ranked third, reflecting some ability to encode visual-semantic regularities despite being trained for image-text alignment rather than geometric consistency. By comparison, SigLIP2, TIPS, and C-RADIOv2 performed substantially worse, indicating that objectives centered on semantic matching or mixed supervision may weaken part-level consistency needed for viewpoint-robust segmentation.

We theorize that the consistent underperformance of VGGT stems from a mismatch between its architecture and the Hummingbird retrieval setting. VGGT is designed as a geometry-grounded, multi-view transformer whose aggregator learns to combine information across several aligned frames of the same object or scene [27]. The original VGGT paper does not include dense segmentation or retrieval-style evaluation; instead, it focuses on 3D geometry tasks such as camera pose, depth, and surface normal estimation. In our evaluation, the model receives only a single view ($S = 1$), meaning the aggregator cannot make use of its multi-view fusion objective. Moreover, VGGT does not provide ViT-style patch embeddings, so we adapt its aggregator outputs to the patch-token interface required by Hummingbird. Together, these factors lead to the weak dense semantic segmentation performance observed.

## 4.2 Experiment B: breaking point analysis

In this experiment, we analyze whether the models experience a sudden failure in viewpoint generalization, and if so, at what angle. The goal is to identify the angular range within which performance remains stable under the Extreme difficulty. In particular, we perform a breaking point analysis under the Extreme setting from experiment A, where only a single reference bin ($0°$) is available in the memory and models are evaluated across the remaining bins. We define the breaking point as the earliest validation bin where a model's normalized performance drops significantly compared to the previous bin. Specifically, we compute the performance drop $\Delta_i$ per bin $i$ as

$$\Delta_i = \text{norm mIoU}_i - \text{norm mIoU}_{i-1}, \tag{2}$$

$$\text{norm mIoU}_i = \frac{\text{mIoU}_i}{\text{mIoU}_{0°}}, \tag{3}$$

where the normalization by the $0°$ bin ensures that performance drops reflect viewpoint sensitivity rather than overall model scale.

A breaking point is recorded at bin $i$ if $\Delta_i \leq -0.1$, indicating a relative drop of 10% or more. This analysis highlights each model's resistance to viewpoint shifts and identifies the angular range within which performance remains stable. We also report the normalized mIoU degradation curves as the viewpoint angle increases.

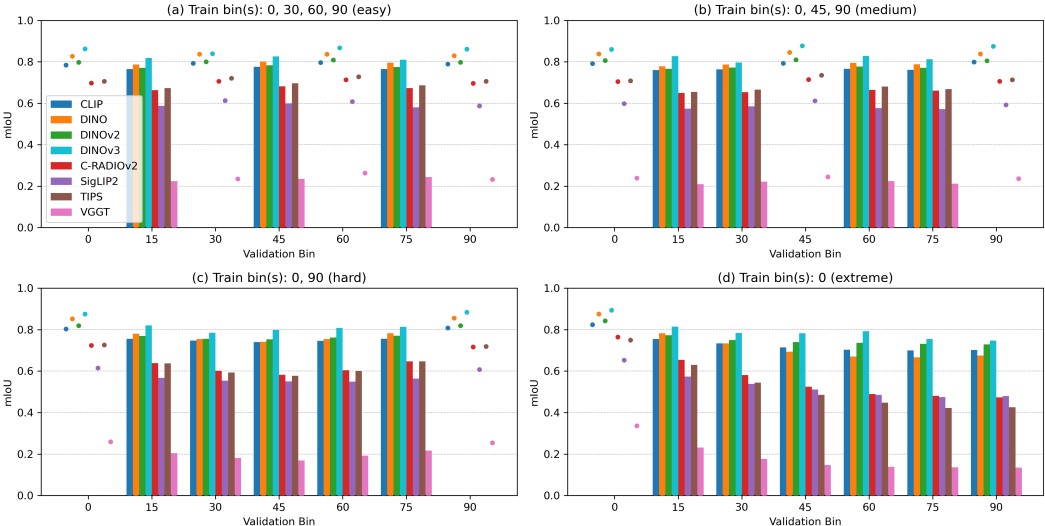

Figure 2: **Segmentation performance across viewpoint bins**. Each subplot (a)–(d) corresponds to a different difficulty level. Bars show mIoU scores on unseen validation bins, while dots show performance on the reference bins stored in the memory bank. The figure illustrates how segmentation accuracy declines as the validation angles move farther from the reference viewpoints. DINOv3, DINOv2, DINO, and CLIP maintain higher, more stable performance as angular distance increases, whereas other models degrade more quickly. VGGT consistently underperforms all other models, yet still follows the expected degradation pattern observed across difficulty levels. Overall, the trend highlights how self-supervised pretraining leads to smoother cross-view generalization.

Table 3: **Breaking points**. For each model in experiment B, we report the breaking point validation bin where the normalized mIoU drop exceeds 10% ($\Delta_i \leq -0.1$) relative to the previous bin. "None" indicates gradual degradation without a sudden breakdown. Only TIPS and C-RADIOv2 reach a breaking point, both at 30°, whereas DINO, DINOv2, CLIP, and SigLIP2 maintain smoother decline curves.

| Model | Breaking point bin | norm. mIoU drop |
|---|---|---|
| CLIP ViT-B/16 | None | – |
| DINO ViT-B/16 | None | – |
| DINOv2 ViT-B/14 | None | – |
| DINOv3 ViT-B/16 | None | – |
| C-RADIOv2 ViT-B/16-CPE | None | – |
| SigLIP2 B/16-512 | None | – |
| TIPS ViT-B/14-HR | 30 | $-0.1148$ |
| VGGT | 30 | $-0.1637$ |

#### 4.2.1 Results

As illustrated in Figure 3, most models show a gradual decline in normalized mIoU as the validation bin angle increases. TIPS and VGGT are the models that reach a breaking point, both at the 30° bin (see Table 3). TIPS has a sharper decrease than C-RADIOv2, with a decrease of 0.115 between 15° and 30°. VGGT's performance drop is significantly larger: it drops by more than 0.2 between 15° and 45°. In contrast, DINOv2, DINOv3, CLIP, DINO, and SigLIP2 degrade smoothly without reaching a breaking point.

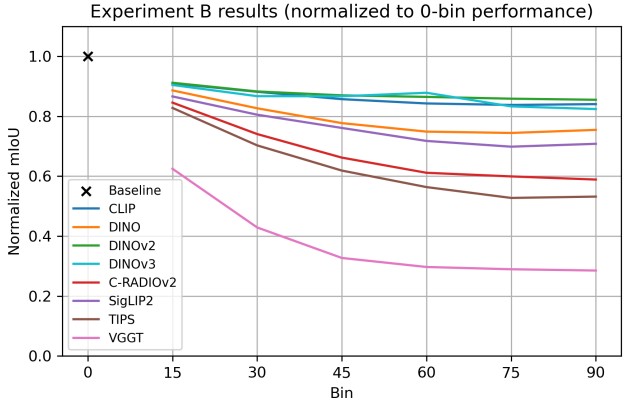

Figure 3: **Normalized mIoU under viewpoint shifts**. Each curve shows the normalized mIoU of a model, measured relative to its 0° reference-bin performance. The horizontal axis represents increasing viewpoint angles (validation bins). A steep drop between consecutive bins corresponds to a potential breaking point in generalization. We can see DINOv2 and DINO degrade more smoothly than TIPS and C-RADIOv2, especially between 15° and 30°. This may indicate that the latter models have weaker viewpoint robustness. DINOv3 peaks at the 60° validation bin, and then drops below the performance of DINOv2 and CLIP, resulting in an inconsistent degradation curve.

### 4.2.2 Discussion

The breaking point analysis highlights differences in robustness across models and shows how their pretraining objectives affect stability under viewpoint shifts. DINOv2 is the most stable, maintaining gradual degradation without sudden drops. CLIP shows a similar trend, with slightly weaker overall performance. DINOv3 follows DINOv2 and CLIP closely and is the only model that shows a slightly fluctuating degradation pattern. Potentially, this is due to architectural factors such as the influence of register tokens and its more complex pretraining objective. Alternatively, it may be affected by the reduced per-bin sample counts noted earlier. More research is needed to clarify the contribution of these factors. The normalized performances of DINO and SigLIP2 also decrease steadily, with a slight increase at the 90° bin. Interestingly, CLIP and SigLIP2 preserve some resilience despite being trained for image-text alignment rather than spatial structure. Although C-RADIOv2 does not reach a breaking point, it comes very close to one, which we could attribute to its mixed supervision weakening the reliability of part-level representations.

In contrast, TIPS and VGGT exhibit clear breakdowns, indicating that their learned features are less robust when faced with unseen viewpoints. For TIPS, this instability likely stems from its focus on semantic alignment rather than spatial consistency. VGGT breaks for a different reason: it is designed for multi-view fusion, and receiving only a single reference view places it outside its intended use. Relative to the baseline, where validation bins are also included in memory, these observations confirm that the sudden declines arise from missing reference views rather than instability in the evaluation pipeline. Overall, the results show that self-supervised encoders such as the DINO-based models produce more geometry-aware features under strong viewpoint shifts, while multimodal or distilled approaches are more brittle. DINOv2's stability makes it the most reliable choice for applications that must handle abrupt perspective changes.

## 4.3 Experiment C: memory size robustness

In this final experiment, we analyze how memory bank size affects viewpoint generalization, comparing the performance gains across difficulty levels against the added computational cost. We follow the Hummingbird paper [1] and vary the number of entries in the memory bank. Three memory sizes are evaluated: 320k, 640k, and 1,024k. For each size, we report segmentation accuracy and measure computational cost, including runtime and resource usage.

Table 4: **Comparison of memory-bank size on segmentation performance**. Mean mIoU scores and standard deviations are shown for three memory sizes ((a) 320k, (b) 640k, (c) 1,024k) and four difficulty levels. Bold values mark the best score per difficulty-memory configuration. Increasing memory generally improves performance, especially for weaker encoders such as SigLIP2. The DINO models remain the top performers across all settings, but have minimal gains past 640k.

| Model | Easy | Medium | Hard | Extreme |
|---|---|---|---|---|
| (a) Memory: 320k | | | | |
| CLIP ViT-B/16 | 0.729 ± 0.138 | 0.725 ± 0.141 | 0.710 ± 0.149 | 0.681 ± 0.156 |
| DINO ViT-B/16 | 0.741 ± 0.147 | 0.736 ± 0.150 | 0.712 ± 0.163 | 0.656 ± 0.178 |
| DINOv2 ViT-B/14 | 0.738 ± 0.145 | 0.736 ± 0.146 | 0.726 ± 0.152 | 0.709 ± 0.158 |
| DINOv3 ViT-B/16 | **0.791 ± 0.162** | **0.790 ± 0.155** | **0.785 ± 0.163** | **0.761 ± 0.187** |
| C-RADIOv2 ViT-B/16-CPE | 0.588 ± 0.136 | 0.579 ± 0.139 | 0.539 ± 0.146 | 0.467 ± 0.149 |
| SigLIP 2 B/16-512 | 0.506 ± 0.154 | 0.501 ± 0.156 | 0.483 ± 0.157 | 0.443 ± 0.149 |
| TIPS ViT-B/14-HR | 0.600 ± 0.161 | 0.590 ± 0.166 | 0.539 ± 0.178 | 0.437 ± 0.175 |
| VGGT-1B | 0.137 ± 0.123 | 0.128 ± 0.114 | 0.108 ± 0.102 | 0.089 ± 0.086 |
| (b) Memory: 640k | | | | |
| CLIP ViT-B/16 | 0.745 ± 0.132 | 0.740 ± 0.136 | 0.727 ± 0.141 | 0.694 ± 0.150 |
| DINO ViT-B/16 | 0.768 ± 0.137 | 0.761 ± 0.141 | 0.736 ± 0.156 | 0.676 ± 0.173 |
| DINOv2 ViT-B/14 | 0.754 ± 0.138 | 0.750 ± 0.142 | 0.740 ± 0.147 | 0.721 ± 0.155 |
| DINOv3 ViT-B/16 | **0.802 ± 0.155** | **0.799 ± 0.148** | **0.790 ± 0.162** | **0.763 ± 0.191** |
| C-RADIOv2 ViT-B/16-CPE | 0.629 ± 0.132 | 0.615 ± 0.135 | 0.572 ± 0.142 | 0.491 ± 0.150 |
| SigLIP 2 B/16-512 | 0.541 ± 0.152 | 0.531 ± 0.154 | 0.511 ± 0.157 | 0.466 ± 0.149 |
| TIPS ViT-B/14-HR | 0.644 ± 0.144 | 0.628 ± 0.153 | 0.571 ± 0.167 | 0.453 ± 0.170 |
| VGGT-1B | 0.168 ± 0.129 | 0.151 ± 0.119 | 0.130 ± 0.102 | 0.103 ± 0.083 |
| (c) Memory: 1,024k | | | | |
| CLIP ViT-B/16 | 0.755 ± 0.130 | 0.748 ± 0.134 | 0.734 ± 0.140 | 0.701 ± 0.149 |
| DINO ViT-B/16 | 0.782 ± 0.132 | 0.774 ± 0.137 | 0.748 ± 0.153 | 0.686 ± 0.171 |
| DINOv2 ViT-B/14 | 0.763 ± 0.136 | 0.758 ± 0.139 | 0.748 ± 0.143 | 0.728 ± 0.154 |
| DINOv3 ViT-B/16 | **0.807 ± 0.153** | **0.805 ± 0.146** | **0.794 ± 0.161** | **0.766 ± 0.191** |
| C-RADIOv2 ViT-B/16-CPE | 0.653 ± 0.129 | 0.636 ± 0.132 | 0.592 ± 0.141 | 0.506 ± 0.150 |
| SigLIP 2 B/16-512 | 0.564 ± 0.150 | 0.551 ± 0.153 | 0.530 ± 0.157 | 0.481 ± 0.152 |
| TIPS ViT-B/14-HR | 0.667 ± 0.137 | 0.647 ± 0.146 | 0.588 ± 0.162 | 0.462 ± 0.169 |
| VGGT-1B | 0.190 ± 0.132 | 0.172 ± 0.119 | 0.146 ± 0.102 | 0.112 ± 0.083 |

### 4.3.1 Results

The results in Table 4 show that increasing memory generally improves mIoU, with the largest gains observed under the Easy difficulty. Doubling memory from 320k to 640k increases performance by about 0.030 mIoU on Easy, while a further increase to 1,024k yields a smaller improvement of 0.017 (see Table 10 in Appendix E.4). DINOv3 achieves the highest absolute performance across all memory sizes, followed by DINO and DINOv2. Weaker models such as C-RADIOv2, SigLIP2, TIPS and VGGT show the greatest relative gains from increasing the memory. All but the former improve by more than 0.048 when moving from 320k to 1,024k, with most of that improvement coming from the first memory doubling.

The runtime analysis in Table 11 in Appendix E.4 shows that larger memory banks increase computational cost. The highest configuration adds about two hours per job. Total memory usage remains similar (189–214 GB), but CPU efficiency declines as memory grows: for example, C-RADIOv2 drops from 16.35% at 320k to 12.52% at 1,024k.

### 4.3.2 Discussion

These results confirm that larger memory banks improve generalization, but the benefits vary across models. Less robust encoders such as C-RADIOv2, SigLIP2, and TIPS gain the most from additional memory, while DINOv3, DINO, and DINOv2 already perform strongly even with smaller banks. The diminishing gains beyond 640k and the higher computational cost suggest that the intermediate configuration offers the best trade-off. This shows that memory scaling can compensate for weaker representations, but stable self-supervised features remain more effective overall.

# 5 Qualitative analysis of segmentations

In addition to quantitative evaluations, we conducted a qualitative inspection of predicted segmentation masks to better understand model behaviour under the large-angle changes of the Extreme difficulty. The illustrations can be found in Appendix E.5.

## 5.1 Localized shape recovery

As illustrated in Figure 28, models generally segment the global shape of an object even when the viewpoint differs substantially from any reference view. Although predictions (center) are not perfectly pixel-aligned with the ground truth (left), the overall contour and structure are typically preserved. Minor over-extensions at the ends of the object (e.g., protruding elements like a tail) are common, reflecting the limits of fine-grained spatial recall under large angular deviations.

## 5.2 Prediction versus ground truth overlays

Overlays of the predicted mask (center) versus the ground truth (left) on the input image, such as those in Figure 29, reveal systematic patterns: models tend to underestimate boundaries in regions with strong self-occlusion or low illumination, while occasionally oversegmenting areas with visually similar textures or shadows. Despite these deviations, the predictions largely align with the main visual cues of the target object, suggesting strong local feature retrieval even from distant viewpoints.

## 5.3 Prediction versus ground truth

Color-coded difference maps (e.g., Figure 29 (right), or Figure 30) show that most misalignments occur near object borders or in thin, high-curvature regions (e.g., legs, tails). This indicates that memory retrieval may struggle with fine-grained spatial resolution under Extreme difficulty views. As illustrated in Figures 7 and 21–26 for the *bed* class, some categories may have imprecise or inconsistent ground truth masks which contribute to measured errors, sometimes penalizing predictions that are visually more plausible. Such cases highlight that annotation quality can limit the reliability of quantitative evaluation, since metrics are tied to imperfect labels rather than perceptual correctness.

## 5.4 Failures

Failure cases, such as the *toy dragon* shown in Figure 30, typically arise when the model retrieves visually similar but geometrically inconsistent patches. This may lead to boundary inflation. Such failures explain the sharp performance drops observed in certain models (e.g., TIPS) at larger angular deviations, where cross-view consistency becomes more challenging.

# 6 Conclusion

We studied the ability of frozen ViT encoders to generalize across unseen camera viewpoints in an in-context segmentation setting, using the Hummingbird architecture and MVImgNet dataset. Our benchmark covered three aspects: controlled evaluation under view shifts, robustness through breaking point analysis, and the effect of memory bank size. Our findings show that self-supervised encoders, particularly DINO, DINOv2, and DINOv3, provide more stable geometry-aware features under viewpoint changes, while multimodal and distilled approaches are more brittle. VGGT underperforms because its multi-view architecture and training objective are likely misaligned with the single-view, patch-based retrieval setup used in our experiments. Larger memory banks help weaker models but offer diminishing returns for stronger encoders, raising questions about the cost-benefit trade-off.

Taken together, these results highlight the strengths and limitations of current ViTs for multi-view perception. While modern self-supervised training yields robust features, none of the models are immune to viewpoint-induced degradation. Addressing this gap will likely require objectives or architectures that enforce 3D consistency. Future work could include extending this evaluation to multi-object or multi-class scenes, wider angular ranges, and compound rotations, providing a broader testbed for building truly viewpoint-robust representations.

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

# A    Licenses for assets

All datasets and models used in this work are released under open licenses. We list them in Table 5 below.

VGGT is released under it's own custom VGGT License (v1 Last Updated: July 29, 2025). Non-commercial use is permitted, allowing academic and research use.

Table 5: **Licenses for datasets and models.** All assets are used in compliance with their respective licenses.

| Asset | License |
|-------|---------|
| MVImgNet | CC BY-NC 4.0 (code), dataset Terms of Use |
| Hummingbird | MIT License |
| CLIP | MIT |
| DINO | Apache 2.0 |
| DINOv2 | Apache 2.0 |
| DINOv3 | DINOv3 License |
| SigLIP2 | Apache 2.0 |
| C-RADIOv2 | NVIDIA Open Model License |
| TIPS | Apache 2.0 (code), CC BY 4.0 (docs) |
| VGGT | VGGT License v1 |

# B    The MVImgNet dataset

The statistics for the reorganized subset of MVImgNet [32] we used are shown in Table 6. The selected subset includes 15 object classes, each represented across 7 standardized angle bins: [0°, 15°, 30°, 45°, 60°, 75°, 90°]. Figure 4 shows an instance of each angle for each object class.

Table 6: **Angle selection accuracy per class**. Mean and standard deviation of angular error (in degrees) for each object class after selecting views closest to the predefined angle bins. Bin image counts marked with † indicate the bin sizes used for the models DINOv3 and VGGT. Due to a misconfiguration, the dataset sizes were reduced for eight classes for these two models.

| Class number | Category | Std. error | Mean error | Images per bin |
|:---:|---|:---:|:---:|:---:|
| 7 | Stove | 1.28 | 0.01 | 197 |
| 8 | Sofa | 1.15 | -0.04 | 91 |
| 19 | Microwave | 1.44 | -0.04 | 120 |
| 46 | Bed | 1.17 | -0.00 | 23 |
| 57 | Toy cat | 1.96 | -0.01 | 783 (762†) |
| 60 | Toy cow | 1.96 | -0.02 | 735 (697†) |
| 70 | Toy dragon | 1.62 | 0.01 | 627 |
| 99 | Coat rack | 1.41 | -0.02 | 97 (9†) |
| 100 | Guitar stand | 1.53 | 0.04 | 218 (3†) |
| 113 | Ceiling lamp | 1.64 | -0.03 | 154 (75†) |
| 125 | Toilet | 1.31 | 0.02 | 58 (5†) |
| 126 | Sink | 1.20 | -0.12 | 30 (5†) |
| 152 | Strings | 1.25 | 0.03 | 192 (121†) |
| 166 | Broccoli | 2.04 | -0.03 | 210 |
| 196 | Durian | 1.65 | 0.03 | 758 |

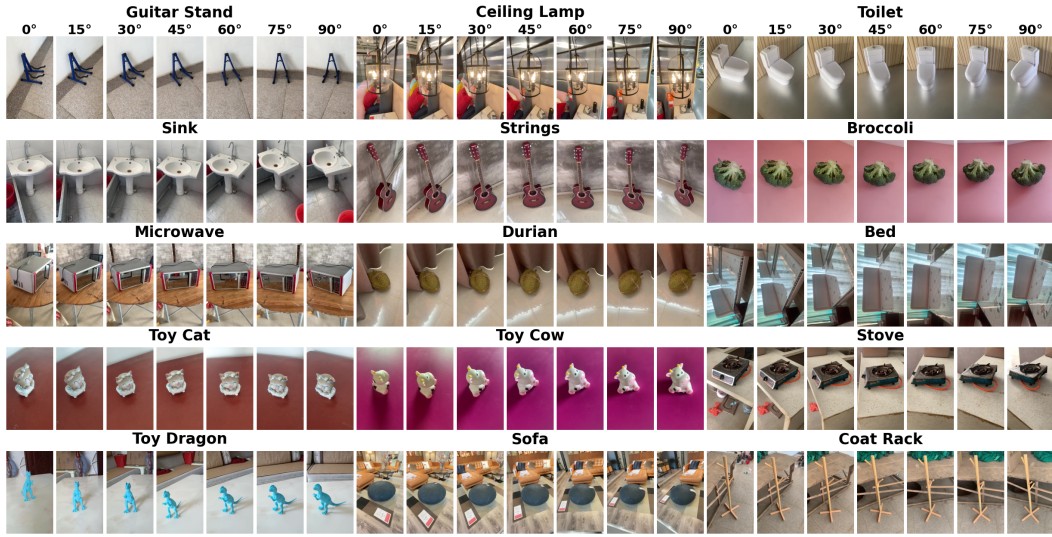

Figure 4: **Multi-view categories**. One representative instance is shown for each of the 15 selected MVImgNet classes across all viewpoint bins (0°–90°). Based on the category taxonomy of MVImgNet [32], we have examples from both the *food* and *artifacts* parents class, which encompass the following children classes: *fruits and vegetables*, *toy dolls* and a series of *instruments*: *furniture*, *kitchen ware*, *home appliances*, and *entertainment*.

## C    Model specifics

The following table details the ViT-based encoders used in our experiments, including their patch configurations and feature dimensionality.

Table 7: **Vision Transformer configurations.** We show the architectural and input settings of the models evaluated in our experiments. Input, patch, and batch sizes are indicated with Input sz., Patch sz., and Batch sz., respectively.

| Model | Architecture | Input sz. | Patch sz. | # patches | Feat. dim. | Batch sz. |
|---|---|---|---|---|---|---|
| DINO | ViT-B/16 | 512 | 16 | 1024 | 768 | 4 |
| DINOv2 | ViT-B/14 | 504 | 14 | 1296 | 768 | 4 |
| DINOv3 | ViT-B/16 | 512 | 16 | 1024 | 768 | 4 |
| OpenAI CLIP | ViT-B/16 | 512 | 16 | 1024 | 768 | 4 |
| C-RADIOv2 | ViT-B/16-CPE | 512 | 16 | 1024 | 768 | 4 |
| SigLIP2 | ViT-B/16-512 | 512 | 16 | 1024 | 768 | 4 |
| TIPS | ViT-B/14-HR | 504 | 14 | 1296 | 768 | 4 |
| VGGT | Geometry-Grounded | 504 | 14 | 1296 | 1024 | 4 |

The models differ in their native input resolutions: $224 \times 224$ for CLIP ViT-B/16 [16], DINO ViT-B/16 [3], DINOv3 ViT-B/16 [24], VGGT [27] and C-RADIOv2 ViT-B/16-CPE [8, 18]; $512 \times 512$ for DINOv2 ViT-B/14 [15]; $504 \times 504$ for SigLIP2 B/16-512 [26]; and $448 \times 448$ for TIPS ViT-B/14-HR [12].

VGGT employs the ViT-L backbone used in DINOv2, consisting of 24 transformer blocks that alternate between frame-wise and global self-attention. Each input image is patchified into tokens using a pretrained DINOv2 [15] tokenizer, maintaining compatibility with other ViT-based encoders in our study. The alternating-attention design allows the encoder to integrate information across frames while refining features within each image. Tokens from selected intermediate blocks are later used by a DPT [17] head for upsampling. Compared to the standard ViT-B encoders (e.g., DINO, CLIP, DINOv2), VGGT's larger capacity and dual-attention structure make it architecturally distinct.

# D Computational resources

Table 8 outlines the specific hardware and software configurations used to execute all experiments, ensuring reproducibility and accounting for memory constraints. All experiments were executed on the Snellius HPC cluster.

Table 8: Experimental hardware and configuration.

| Category | Component/value | Notes |
|---|---|---|
| **Hardware** | $4 \times$ A100 GPUs | High-performance cluster |
| | 72 CPU cores | |
| | Max wall time | 30 hours per job |
| **Software/settings** | Search index distribution | FAISS sharding across GPUs |
| | Batch size | 4 |
| | Random seed | Python, NumPy, PyTorch, CUDA (set to 42) |

# E Additional results

## E.1 Reproduction

Table 9: **Reproduction results.** We report accuracy (%) at different evaluation scales. "Reported" indicates values taken from prior published results, while "Reproduced" refers to our reproduction with a batch size (BS). Bold values highlight the best result for each model-scale configuration. Overall, our reproduced scores are close to or exceed the reported results in the original paper, with reproduced results differing by less than 3%.

| Model | Source | $1024 \times 10^2$ | $1024 \times 10^3$ | $1024 \times 10^4$ |
|---|---|---|---|---|
| ViT-S/16 | Reproduced, BS: 256 | 37.5 | 45.0 | – |
| | Reproduced, BS: 64 | **37.9** | **45.1** | **49.3** |
| | Reported, BS: 64 | 37.2 | 43.1 | 46.6 |
| ViT-B/16 | Reproduced, BS: 64 | **48.0** | **54.7** | – |
| | Reproduced, BS: 32 | 47.8 | 54.6 | – |
| | Reproduced, BS: 8 | – | – | **57.9** |
| | Reported, BS: 64 | 44.9 | 50.8 | 55.7 |
| ViT-S/14 | Reproduced, BS: 64 | 69.6 | **75.1** | **77.0** |
| | Reported, BS: 64 | **70.2** | 74.9 | 77.0 |
| ViT-B/14 | Reproduced, BS: 64 | 68.0 | 74.0 | – |
| | Reproduced, BS: 8 | – | – | **76.6** |
| | Reported, BS: 64 | **69.1** | **74.6** | 76.9 |
| ViT-L/14 | Reproduced, BS: 64 | 64.1 | 71.2 | – |
| | Reproduced, BS: 8 | – | – | 74.4 |
| | Reported, BS: 64 | **64.6** | **71.7** | **74.8** |
| ViT-G/14 | Reproduced, BS: 32 | **62.4** | – | – |
| | Reproduced, BS: 16 | – | **70.1** | – |
| | Reproduced, BS: 8 | – | – | 73.3 |
| | Reported, BS: 64 | 62.3 | 69.9 | **73.6** |

## E.2 Experiment A

In experiment A, we assess model generalization across view-angle difficulties using per-class performance curves. In the figures below, the bars represent mean mIoU for unseen validation bins, and dots mark reference viewpoints. Performance typically declines as viewpoint distance increases from the reference bins, consistent with the trend in Figure 2. Across viewpoint difficulties, DINO, DINOv2, and CLIP consistently outperform all other models. C-RADIO follows closely, while SigLIP2 and especially TIPS degrade substantially under extreme viewpoints (with TIPS showing the weakest overall robustness). DINOv3 generally outperforms all models; however, this does not hold for the *coat rack* and *guitar stand* classes, where, in the Extreme difficulty scenario, it becomes one of the worst-performing models. This drop may be influenced by the reduced per-bin image counts (as noted in Table 6), making the class results less stable.

While the other models exhibit consistent viewpoint robustness, the VGGT-1B model behaves markedly differently. Its mIoU remains low across all classes and difficulty levels, typically below 0.4, with only minor variation between bins. This is expected given the architectural mismatch with Hummingbird: VGGT is a geometry-grounded, multi-view transformer, and in our setting, it receives only a single view and must rely on adapted aggregator tokens rather than native patch embeddings. Under these conditions, its predictions often collapse toward background-heavy masks, yielding high background mIoU but near-zero scores for small or fine-grained objects. As a result, its degradation curve appears shallow, not because the model is stable, but because performance stays uniformly low in this adapted, single-view configuration.

Across all models, the background class performs best; other classes that perform well are *durian*, *broccoli*, and *strings*. These tend to generalize above average under unseen viewpoints. In contrast, *bed*, *coat rack*, *guitar stand*, and *sink* perform worst (the largest negative generalization gaps). For the *sink* class, DINOv3 significantly outperforms all other models. For *bed*, the drop is explained by poor ground-truth annotations that likely reduce model scores (as explored in Appendix E.2.1).

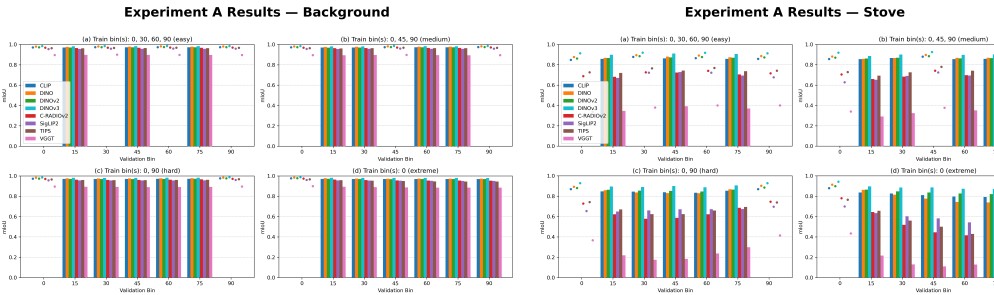

Figure 5: **Experiment A: background**. Viewpoint generalization for the background class across four training difficulty settings: (a) easy, (b) medium, (c) hard, and (d) extreme. All models achieve near-perfect consistency across bins, indicating that the background segmentation task remains invariant to viewpoint changes. Since VGGT scores highly only on background while remaining low on all object classes, this discrepancy indicates that the model may over-predict background regions, leading to inflated background mIoU despite poor object segmentation.

Figure 6: **Experiment A: stove**. The stove class perform slightly above the dataset average shown in in Figure 2. The mIoU remains consistently high, and relative difference stays positive across all models. SigLIP2 and CLIP show the strongest viewpoint robustness. Degradation across validation bins follows the general pattern of Figure 2, with no major model-specific failures.

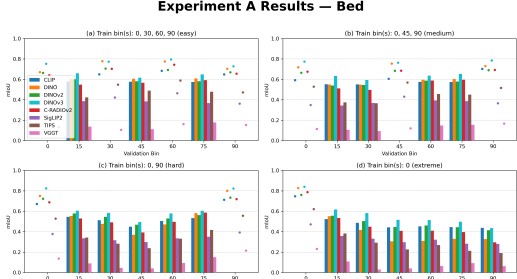

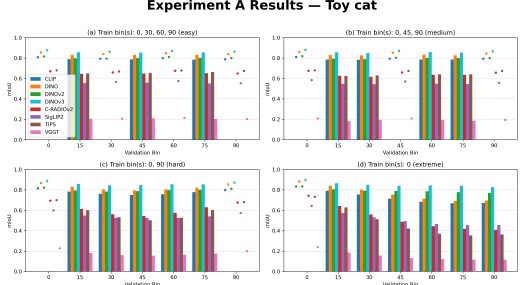

Figure 7: **Experiment A: bed**. The bed class scores far below the all-class baseline, with it's mIoU scores remaining low even for strong models. This degradation is largely explained by systematically incorrect ground-truth masks, as seen in Figures 21–26 we analyze in Appendix E.2.1. The figure reflects this annotation-driven performance drop.

Figure 8: **Experiment A: toy cat**. The toy cat performance is very similar to the dataset-wide average observed in Figure 2, with relative differences near zero. Strong models (CLIP, DINO) achieve high mIoU (0.77–0.84) and accuracy drops consistently across difficulty settings.

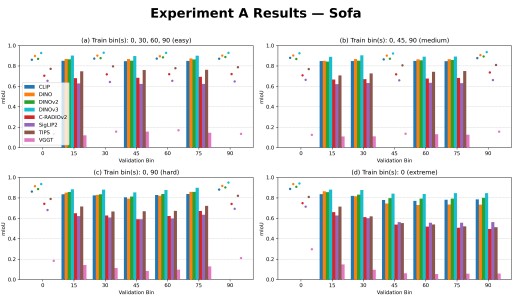

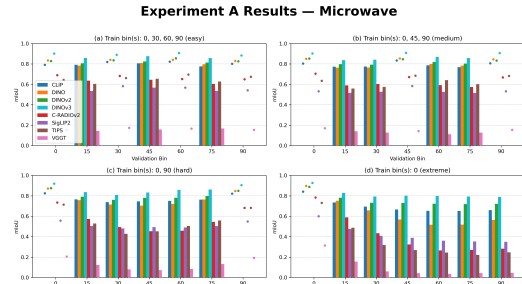

Figure 9: **Experiment A: sofa**. The sofa class performs slightly above the overall class average for every model and difficulty setting. The mIoU performance decreases smoothly with increased viewpoint difficulty, matching the overall experiment A trend observed in Figure 2.

Figure 10: **Experiment A: microwave**. This class underperforms relative to Figure 2, with the largest drops appearing under Hard and Extreme difficulty. Model behaviour diverges notably: DINOv3, DINOv2 and CLIP maintain comparatively strong performance, whereas DINO shows a mild decline. TIPS and SigLIP2 appear to struggle when segmenting the microwave category and degrade sharply under the Extreme condition, with a pronounced collapse beyond the 45° bin. Despite these weaker generalization patterns, the overall degradation across viewpoint bins follows the expected trend.

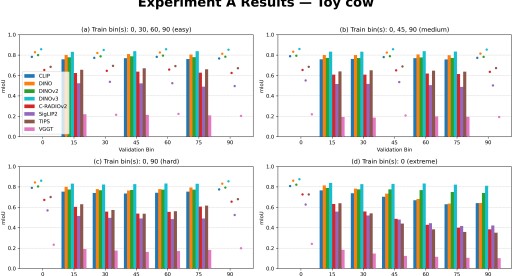

Figure 11: **Experiment A: toy cow**. The toy cow category shows slightly lower performance across all models on unseen bins, placing it among the weaker classes. Most noticeable is the lower performance of SIGLIP2 in the easy and medium task (see for example angle bin 75° in the easy task).

Figure 12: **Experiment A: toy dragon**. This class exhibits mildly negative gaps, in particular C-RADIO is always lower than TIPS (when comparing to Figure 2). As a whole though, its performance trends mirror the general degradation observed of weaker models struggling as viewpoint distance increases.

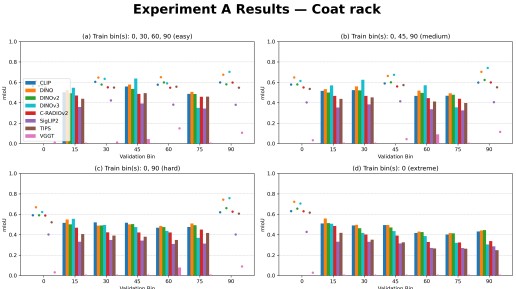

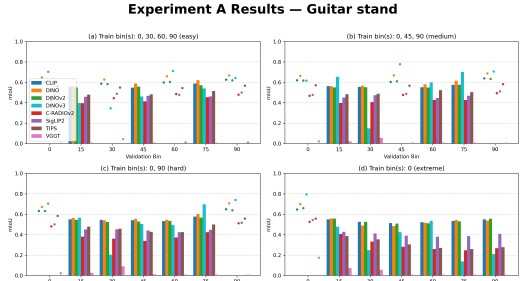

Figure 13: **Experiment A: coat rack**. The coat rack is the worst-performing class overall, with mIoU remaining below 0.7 across all validation bins. VGGT performs particularly poorly, often producing near-zero mIoU. This likely indicates a failure to localize the object rather than viewpoint sensitivity. As seen in Figure 4, the object's structure is extremely thin, which likely makes it highly sensitive to viewpoint changes and contributes to its weak generalization. It remains an open question whether this difficulty is intrinsic to the geometry or arises from dataset-specific factors. This could be an area of further investigation.

Figure 14: **Experiment A: guitar stand**. The Guitar stand class scores well below the baseline for all models and difficulties. Degradation is steep: DINOv3 consistently struggles with the 30° validation bin, performs very poorly in the Extreme task, and generally appears to have a large negative gaps (drops by more than 0.2 mIoU) when compared to the "all-class" baseline. VGGT once again produces near-zero mIoU. Like the coat rack, the thin structure likely contributes to poor performance. However, exact cause remains unclear and is a point for future investigation.

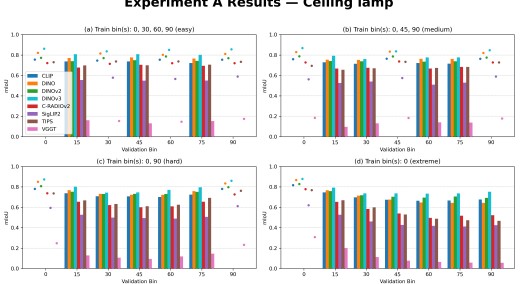

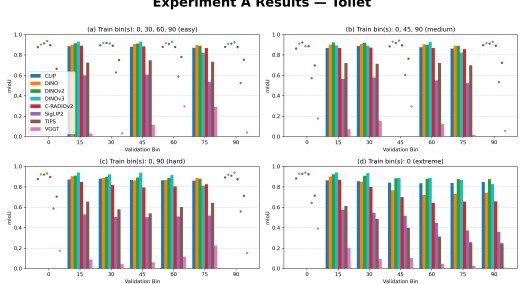

Figure 15: **Experiment A: ceiling lamp**. Performance trends closely mirror the general degradation observed in Figure 2, indicting the class has average viewpoint robustness. Degradation across bins follows the expected trend, with no unusual model failures.

Figure 16: **Experiment A: toilet**. The toilet class consistently performs above the overall trend in Figure 2, with C-RADIO showing the strongest boost, achieving increases of nearly 0.2 mIoU across difficulty levels.

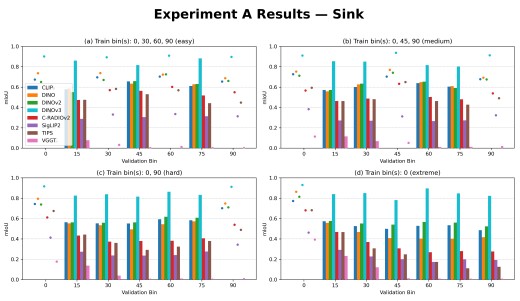

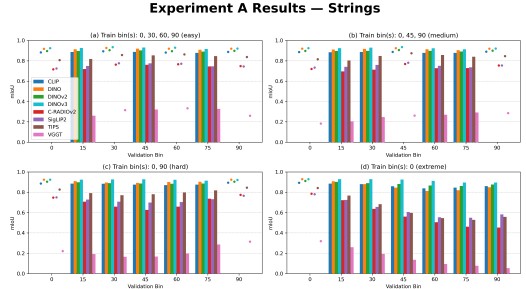

Figure 17: **Experiment A: sink**. The sink class is one one of the worst three class performances in the dataset. DINOv3 is the only model which performs welll in this category. All other models show large negative gaps relative to Figure 2, with SigLIP2 already underperforming in the easy setting by roughly 0.2 mIoU. Absolute scores generally remain low across all difficulty levels, and the degradation with viewpoint variation is steep. What geometric or appearance factors make sinks perform poorly is not immediately obvious, suggesting that sink segmentation is a broader failure mode worth investigating.

Figure 18: **Experiment A: strings.** The strings class generalises reliably across all models, with performance declining smoothly as viewpoint difficulty increases yet remaining consistently above the dataset average. Since this category spans multiple instruments (e.g. guitar, ukulele, guzheng), future work could investigate which visual attributes they share which lead to strong viewpoint stability.

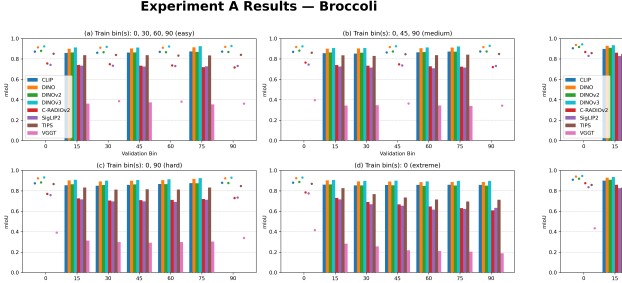

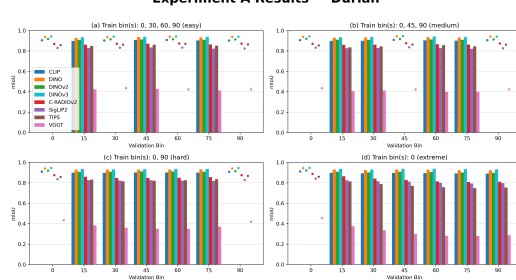

Figure 19: **Experiment A: broccoli.** Models generalize well on the broccoli class, maintaining stable and above-average performance across unseen viewpoint bins. DINOv3 is consistently the top performer, the other DINO-based models and CLIP following closely. TIPS also performs strongly, trailing just behind the leaders in all but the Extreme setting and achieving notably higher scores than in Figure 2. The results invite further investigation into which aspects of broccoli's geometry support such robust viewpoint generalization.

Figure 20: **Experiment A: durian.**. Durian is among the best-performing classes, showing high absolute mIoU (0.78–0.94) across most models and all difficulty levels. Despite VGGT underperforming other models, its performance in this category is among its best, which reinforces this class's easy difficulty. This robustness is somewhat surprising: although the durian has a rich surface texture, its overall geometry is nearly spherical. It remains unclear whether the stable segmentation under viewpoint shifts stems from its distinctive visual signature or from contextual cues in the surrounding scene, such as the curtains highlighted in Figure 4.

### E.2.1 Bed category ground truth analysis

We investigated the poor performance of the *bed* class (as seen in Figure 7) and identified inconsistent or incomplete ground truth annotations as a likely cause.

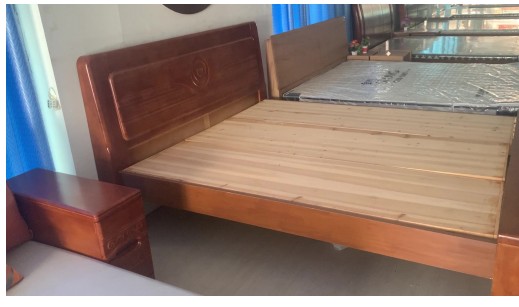

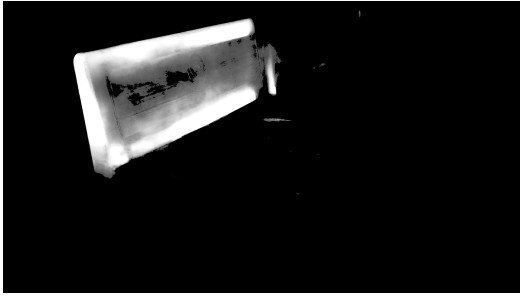

Figure 21: **Bed image example 1.** Original image used for annotation.

Figure 22: **Incomplete annotation.** Ground truth excludes parts of the bed (side railing, footboard) and ignores surrounding beds.

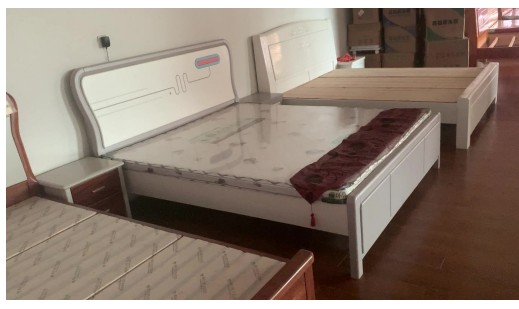

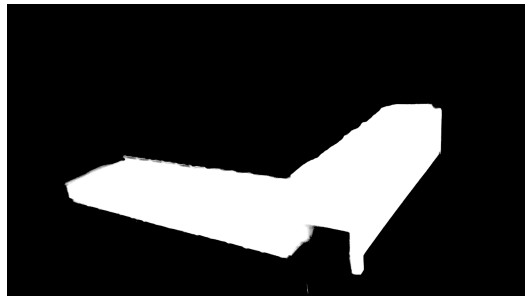

Figure 23: **Bed image example 2.** Original image used for annotation.

Figure 24: **Erroneous annotation.** The ground truth excludes large parts of the bed (headboard, mattress) and incorrectly includes the shadow between two beds.

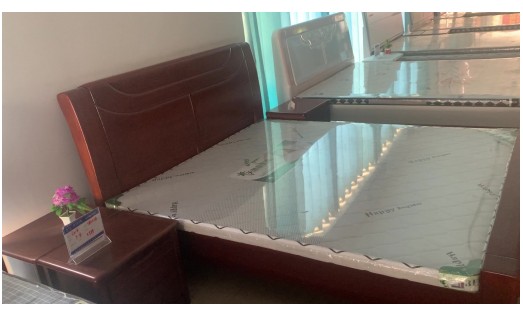

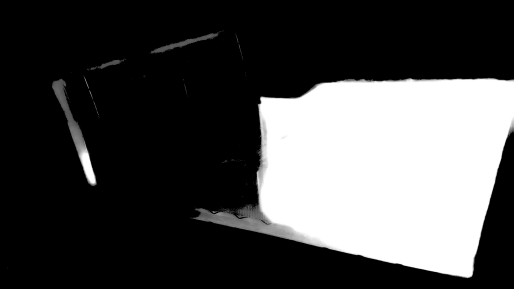

Figure 25: **Bed image example 3.** Original image used for annotation.

Figure 26: **Annotation error.** The ground truth omits the surrounding beds, incorrectly includes the gap between the main bed and the bed to its right, and only partially marks the headboard (side only).

## E.3 Experiment B

As shown in Figure 3 and Figure 27, most models exhibit a gradual performance decline as the distance of the validation bin from the training bin (0°) increases. Visually, only C-RADIOv2 (ViT-B/16-CPE), TIPS (ViT-B/14-HR) and VGGT exhibit sharp performance drops with increasing validation angles, indicating limited generalization capacity under extreme viewpoint shifts.

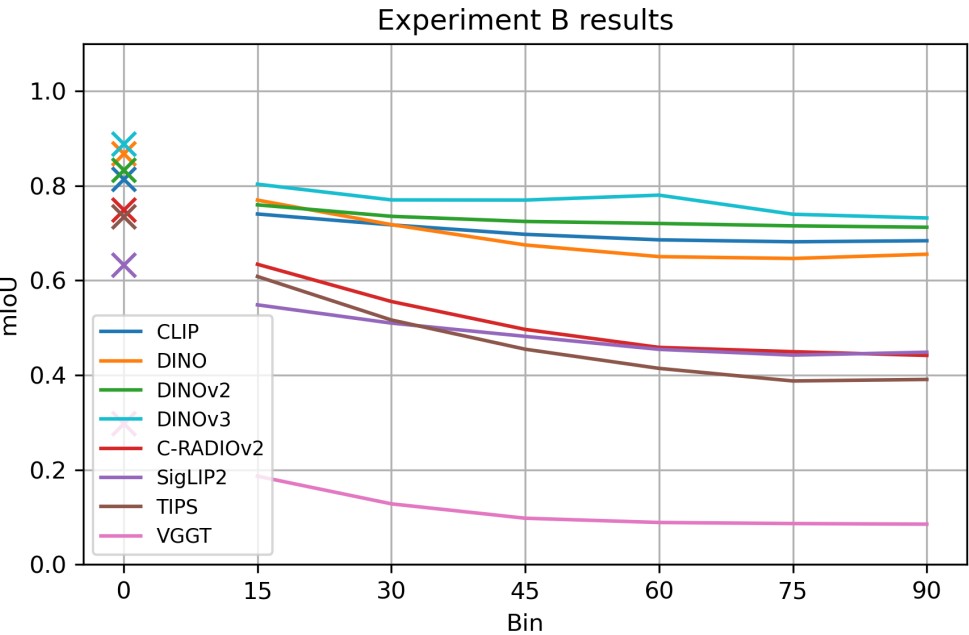

Figure 27: **Absolute mIoU under viewpoint shifts**. Each curve shows the mIoU of a model evaluated on increasing validation bins, measured directly (without normalization to the 0° baseline). The horizontal axis represents viewpoint angle relative to the training bin (0°), and the vertical axis shows the segmentation performance via mIoU. All models show a gradual decrease as the angular distance grows, reflecting reduced correspondence between memory and query views. DINOv2 has the most consistent generalization across bins, maintaining a high mIoU across all bins. DINOv3 maintains the highest mIoU, but has an unexpected peak at bin 60. Further research could explore whether this is attributed to DINOv3's architecture (e.g. its use of register tokens and its more complex pretraining objective), or if it's a result of the reduced per-bin image counts. By contrast, TIPS displays the steepest decline (dropping by more than 0.2 between 15 and 75), confirming its reduced robustness under large viewpoint changes.

Table 10: **Performance gains from memory.** We report absolute mIoU improvements when increasing memory from (a) 320k to 640k, (b) 640k to 1,024k, and (c) 320k to 1,024k. Values are computed as differences from Table 4, with the final column showing the average gain across all four difficulty levels.

| Model | Gains | | | | |
| | Easy | Medium | Hard | Extreme | Average |
|---|---|---|---|---|---|
| (a) Memory: 320k → 640k | | | | | |
| CLIP ViT-B/16 | 0.016 | 0.015 | 0.017 | 0.013 | 0.01525 |
| DINO ViT-B/16 | 0.027 | 0.025 | 0.024 | 0.020 | 0.02400 |
| DINOv2 ViT-B/14 | 0.016 | 0.014 | 0.014 | 0.012 | 0.01400 |
| DINOv3 ViT-B/16 | 0.011 | 0.009 | 0.005 | 0.002 | 0.00675 |
| C-RADIOv2 ViT-B/16-CPE | 0.041 | 0.036 | **0.033** | **0.024** | **0.03350** |
| SigLIP 2 B/16-512 | 0.035 | 0.030 | 0.028 | 0.023 | 0.02900 |
| TIPS ViT-B/14-HR | **0.044** | **0.038** | 0.032 | 0.016 | 0.03250 |
| VGGT | 0.031 | 0.023 | 0.016 | 0.014 | 0.02250 |
| **Average per task** | **0.028** | 0.024 | 0.021 | 0.016 | |
| (b) Memory: 640k → 1,024k | | | | | |
| CLIP ViT-B/16 | 0.010 | 0.008 | 0.007 | 0.007 | 0.00800 |
| DINO ViT-B/16 | 0.014 | 0.013 | 0.012 | 0.010 | 0.01225 |
| DINOv2 ViT-B/14 | 0.009 | 0.008 | 0.008 | 0.007 | 0.00800 |
| DINOv3 ViT-B/16 | 0.005 | 0.006 | 0.004 | 0.003 | 0.0045 |
| C-RADIOv2 ViT-B/16-CPE | **0.024** | **0.021** | **0.020** | **0.015** | **0.02000** |
| SigLIP 2 B/16-512 | 0.023 | 0.020 | 0.019 | **0.015** | 0.01925 |
| TIPS ViT-B/14-HR | 0.023 | 0.019 | 0.017 | 0.009 | 0.01700 |
| VGGT | 0.022 | 0.021 | 0.016 | 0.009 | 0.0170 |
| **Average per task** | **0.016** | 0.015 | 0.0129 | 0.0094 | |
| (c) Memory: 320k → 1,024k | | | | | |
| CLIP ViT-B/16 | 0.026 | 0.023 | 0.024 | 0.020 | 0.02325 |
| DINO ViT-B/16 | 0.041 | 0.038 | 0.036 | 0.030 | 0.03625 |
| DINOv2 ViT-B/14 | 0.025 | 0.022 | 0.022 | 0.019 | 0.02200 |
| DINOv3 ViT-B/16 | 0.016 | 0.015 | 0.009 | 0.005 | 0.01125 |
| C-RADIOv2 ViT-B/16-CPE | 0.065 | **0.057** | **0.053** | **0.039** | **0.05350** |
| SigLIP 2 B/16-512 | 0.058 | 0.050 | 0.047 | 0.038 | 0.04825 |
| TIPS ViT-B/14-HR | **0.067** | **0.057** | 0.049 | 0.025 | 0.04950 |
| VGGT | 0.053 | 0.044 | 0.038 | 0.023 | 0.0395 |
| **Average per task** | **0.0439** | 0.0383 | 0.0348 | 0.0249 | |

## E.4 Experiment C

In experiment C, we evaluate the impact of memory bank size on model performance. Table 10 reports the absolute mIoU gains per difficulty level when increasing the memory bank for enteries (a) from 320k to 640k, (b) from 640k to 1,024k, and (c) from 320k to 1,024k. This highlights how memory sensitivity varies across model architectures. On the other hand, Table 11 outlines the computational trade-offs. We report the runtime and memory usage associated with each model under both memory configurations. This provides context for the computational cost underlying the performance results.

Table 11: **System performance under memory scaling.** We report wall-clock time, memory usage, CPU usage and efficiency, and memory usage and efficiency for viewpoint-based segmentation with memory sizes (a) 320k, (b) 640k, and (c) 1,024k. Some entries are marked as (running) or show zeroed usage values; these correspond to cases where monitoring logs did not update correctly, even though the jobs completed successfully. Bold values highlight the largest drops in CPU efficiency for each model.

| Model | Wall-clock time | CPU | | Memory | |
|---|---|---|---|---|---|
| | | Used | Eff. | Used (GB) | Eff. |
| (a) Memory: 320k | | | | | |
| CLIP ViT-B/16 | 07:32:14 | 3-15:33:05 | 16.13% | 198.14 | 41.28% |
| DINO ViT-B/16 | 07:32:35 | 3-12:20:38 | 15.53% | 214.11 | 44.61% |
| DINOv2 ViT-B/14 | 08:39:50 (running) | 00:00:00 | 0.00% | 0.00 | 0.00% |
| DINOv3 ViT-B/16 | 06:48:47 | 00:00:03 | 0.00% | 185.82 | 38.71% |
| C-RADIOv2 ViT-B/16-CPE | 07:27:39 | 3-15:50:23 | 16.35% | 199.82 | 41.63% |
| SigLIP 2 B/16-512 | 07:39:28 | 3-19:58:28 | 16.68% | 189.89 | 39.56% |
| TIPS ViT-B/14-HR | 08:33:54 (running) | 00:00:00 | 0.00% | 0.00 | 0.00% |
| VGGT | 10:21:59 | 00:00:03 | 0.00% | 188.59 | 39.29% |
| (b) Memory: 640k | | | | | |
| CLIP ViT-B/16 | 08:21:00 | 3-10:39:36 | 13.75% | 177.58 | 37.00% |
| DINO ViT-B/16 | 08:28:26 (running) | 00:00:00 | 0.00% | 0.00 | 0.00% |
| DINOv2 ViT-B/14 | 09:43:52 (running) | 00:00:00 | 0.00% | 0.00 | 0.00% |
| DINOv3 ViT-B/16 | 07:23:44 | 00:00:03 | 0.00% | 199.28 | 41.52% |
| C-RADIOv2 ViT-B/16-CPE | 08:18:48 | 3-11:02:33 | 13.87% | 203.63 | 42.42% |
| SigLIP 2 B/16-512 | 08:26:16 | 3-16:01:55 | 14.49% | 181.39 | 37.79% |
| TIPS ViT-B/14-HR | 09:33:10 (running) | 3-09:16:27 | 11.82% | 209.87 | 43.72% |
| VGGT | 13:00:18 | 00:00:03 | 0.00% | 167.43 | 34.88% |
| (c) Memory: 1,024k | | | | | |
| CLIP ViT-B/16 | 09:20:31 | 3-07:03:04 | 11.75% | 207.41 | 43.21% |
| DINO ViT-B/16 | 09:29:42 | 3-07:07:59 | 11.58% | 213.23 | 44.42% |
| DINOv2 ViT-B/14 | 11:10:37 | 3-11:36:03 | 10.39% | 209.23 | 43.59% |
| DINOv3 ViT-B/16 | 08:22:03 | 00:00:03 | 0.00% | 195.13 | 40.65% |
| C-RADIOv2 ViT-B/16-CPE | 09:28:53 | 3-13:29:38 | 12.52% | 200.34 | 41.74% |
| SigLIP 2 B/16-512 | 09:36:48 (running) | 00:00:00 | 0.00% | 0.00 | 0.00% |
| TIPS ViT-B/14-HR | 11:04:09 | 3-11:57:18 | 10.53% | 211.38 | 44.04% |
| VGGT | 16:07:11 | 00:00:03 | 0.00% | 167.04 | 34.80% |

## E.5 Qualitative analysis

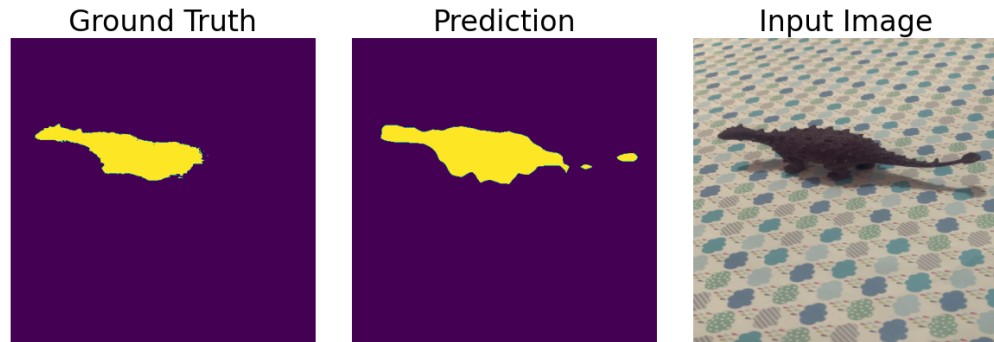

Figure 28: **Qualitative segmentation.** Results shown using DINO. Left: input image. Center: predicted mask. Right: ground truth mask. Interestingly, the prediction of the *toy dragon* aligns more closely with visible object boundaries than the ground truth, which appears coarser and less accurate at fine details (e.g., tail, horns, stomach occlusion).

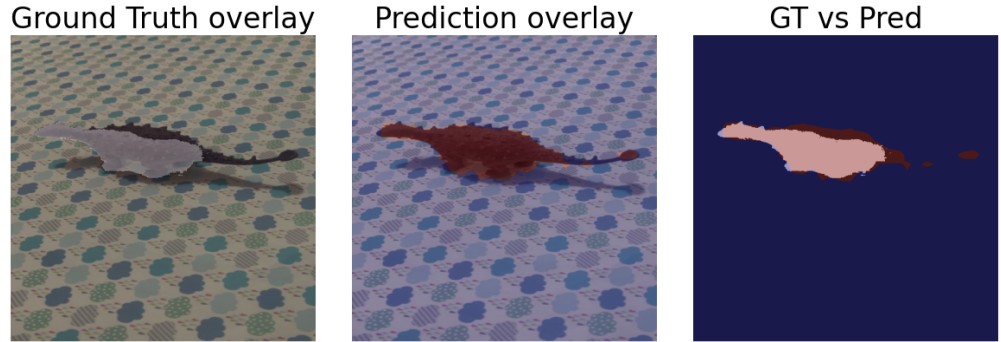

Figure 29: **Overlay comparison of ground truth and prediction.** The same instance as in Figure 28 is shown for better. Left: ground truth overlaid on the input image. Center: prediction overlaid on the input image. Right: difference visualization, with the ground truth overlaid on the predicted mask.

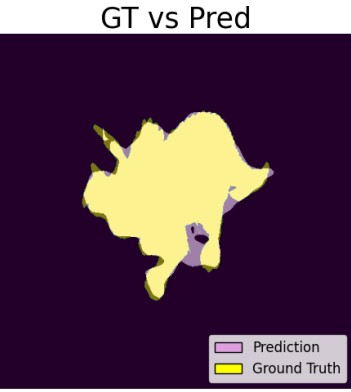

Figure 30: **Another overlay comparison of ground truth and prediction.** Shown using DINO. Yellow: ground truth mask. Purple: predicted mask. We note that discrepancies appear at fine-grained boundaries (e.g., tail, horns, nose) and in occluded or shadowed regions (e.g., under the stomach).

