# OpenReview forum: "Evaluating Foundation Models' 3D Understanding Through Multi-View Correspondence Analysis"
_NeurIPS.cc/2025/Workshop/UniReps — UniReps2025_

### Official Review · Reviewer_zKu7 · 2025-09-04
**Interesting analysis with potential use-cases in autonomous driving**

**Confidence:** 3

**Review:**

# Summary
The paper extends the Hummingbird method (a memory augmented In Context Learning method) to multi-view correspondence and 3D understanding evaluation of foundation models. The authors analyze the zero-shot generalization to novel view segmentation by gradually restricting access to the views available in memory.

# Strengths and Weaknesses
1. The authors propose a novel binning method to save only certain views of an image and show that most of the foundation models can extrapolate to segmenting objects' images taken in novel-views.
2. A detailed analysis of existing ViT based vision encoders exposes their strengths and weaknesses in this extrapolation.
3. The memory-based technique of hummingbird along with the multi-view representation understanding presents a relatively sound fit for the scope of this workshop.
4. An interesting comparison would be to just use the FMs without any memory augmentation, by training a "view angle token" in the ViT's context, as is done in some MV2D-3D models (https://arxiv.org/html/2411.07135v1), as a good baseline for comparison.

**Score:**

4

**Topic Fit:**

2

---

### Official Review · Reviewer_a2o1 · 2025-09-14
**A careful benchmark study of viewpoint robustness in ViTs, though its novelty and motivation remain limited.**

**Confidence:** 4

**Review:**

### Summary

This paper presents an evaluation of multiple popular Vision Transformer encoders in terms of their ability for in-context-learning of image segmentation under different camera angles.
For this purpose, the authors use the Hummingbird framework [1] with the MVImg Dataset that contains different viewpoints of common objects. The authors conduct three experiments, measuring robustness to the available reference view angles, breaking point analysis, and robustness to the context / memory size.
The results reveal that the DINO family of models are the most robust to viewpoint changes, producing segmentations of the highest quality.

### Strengths
1. Geometry awareness and robustness to 3D shifts is an important and timely topic.
2. The paper contributes to the growing body of work on 3D-awareness benchmarks with a novel metric.
3. The paper conducts a very detailed study of the proposed topic, describing all the steps for reproducibility.
4. The authors included their codebase as a part of submission.

### Weaknesses

1. While it is true that “no benchmark currently evaluates a model’s ability to generalize segmentation across viewpoint shifts using dynamic memory” (L52-53), the authors do not justify what shortcomings of previous benchmarks their evaluation addresses, leaving me wondering about its practicality.
2. The benchmark misses several popular encoders that were developed precisely with 3D-awareness in mind, such as VGGT [2], SPA [3], and CroCo [4]. I believe including them would show an interesting comparison of general-purpose encoders vs models geared towards 3D tasks specifically.
3. It seems that “cross-viewpoint” (and not “model”) generalization would be a more appropriate name for section 5
4. The paper’s structure could be improved. For example, Experiments A, B, and C should probably be grouped under a single “Experimental results” section.
5. In my opinion, the main text of the paper is overly detailed with the aspects like the number of CPUs on the author’s machine, references to the codebase, or the job time limit on the author’s cluster. While these can increase reproducibility and help understand the experimental environment, such details distract from the main topic of the paper and can be very well relegated to the appendix.

### Justification of the verdict

While the paper contributes a novel evaluation, it does not justify why this benchmark is needed beyond existing evaluations. Ultimately, the main experimental takeaways (DINOv2 model is the best under viewpoint changes) confirm what many in the community would expect, making the contribution incremental.
Moreover, while the technical details help with reproducibility, they would fit better in an appendix, to improve readability.


[1] Towards in-context scene understanding
Ivana Balaževíc, David Steiner, Nikhil Parthasarathy, Relja Arandjelovic, and Olivier J. Hénaff.
https://arxiv.org/abs/2306.28401667.285

[2] VGGT: Visual Geometry Grounded Transformer
Jianyuan Wang, Minghao Chen, Nikita Karaev, Andrea Vedaldi, Christian Rupprecht, David Novotny
https://arxiv.org/abs/2503.11651

[3] SPA: 3D Spatial-Awareness Enables Effective Embodied Representation
Haoyi Zhu, Honghui Yang, Yating Wang, Jiange Yang, Limin Wang, Tong He
https://openreview.net/forum?id=6TLdqAZgzn

[4] CroCo: Self-Supervised Pre-training for 3D Vision Tasks by Cross-View Completion
Philippe Weinzaepfel, Vincent Leroy, Thomas Lucas, Romain Brégier, Yohann Cabon, Vaibhav Arora, Leonid Antsfeld, Boris Chidlovskii, Gabriela Csurka, Jérôme Revaud
https://arxiv.org/abs/2210.10716

**Score:**

2

**Topic Fit:**

2

---

### Official Review · Reviewer_6mnq · 2025-09-15
**Interesting idea, but missing key comparisons with prior work**

**Confidence:** 4

**Review:**

This paper evaluates how foundation model image encoders process in-context object segmentation under unseen camera angles. Authors curated a dataset by grouping object views and constructed dynamic memory banks from selected viewpoints. Experiments compare generalization by evaluating six pretrained ViT models and prove the benefits of contrastive pretraining for robust performance across large viewpoint shifts

Strengths:
1. The analysis of in-context object segmentation generalization of Vision encoder across viewpoint shift is an interesting topic to the academic community.
2. Better performance with increasing memory bank design.

Weakness：
1. The contribution of the paper is limited by extending the Hummingbird framework with MVImgNet.
2. The experiment analysis is not convincing enough (cross-model generalization, breaking point analysis, memory size robustness). Because the comparison experiment (6 ViT vision encoders) seems to be not enough.
3. Paper is difficult to follow, e.g paragraph logics needs to be improved.

**Score:**

2

**Topic Fit:**

3

---

### Official Review · Reviewer_Fwbe · 2025-09-16
**This paper extends the Hummingbird framework to evaluate foundation models' 3D understanding using MVImgNet, showing DINO-based encoders perform best on cross-view segmentation.**

**Confidence:** 4

**Review:**

The methodology for binning viewpoints based on relative angular deviation from COLMAP extrinsics is technically solid and allows for a controlled assessment of generalization. The angular deviation formula is appropriately used to quantify rotation differences, and selecting one representative frame per bin minimizes variance while maintaining dataset manageability. However, the curation criteria, limiting to 1-6 GB category sizes and requiring full angular coverage across seven bins, introduce potential bias toward simpler objects, as evidenced by the exclusion of categories like laptops. A sensitivity analysis on angular selection errors (mean ~0-0.12 degrees per class, Table 5) would strengthen claims about bin accuracy. Qualitatively, visualizations (Figs. 27-29) reveal boundary errors and over-segmentation in extreme views, partly due to MVImgNet annotation flaws (e.g., incomplete bed masks, Figs. 20-25). This is a fair critique, but quantifying error types (e.g., occlusion vs. lighting) via breakdown metrics would add depth.

**Score:**

4

**Topic Fit:**

3